Pronounced and prevalent intersexuality does not impede the ‘Demon Shrimp’ invasion

Green Etxabe Amaia 1
Short Stephen 1
Flood Tim 2
Johns Tim 2
Ford Alex T. 1 alex.ford@port.ac.uk
1 Institute of Marine Sciences, School of Biological Sciences, University of Portsmouth , Portsmouth , United Kingdom
2 Environment Agency , Howbery Park, Wallingford, Oxfordshire , UK
De Baets Kenneth
Electronic publication date: 2015 Feb 10
Publication date: 2015
Volume: 3
Electronic Location ID: e757
Received 2014 Dec 10; Accepted 2015 Jan 21
Copyright: © 2015 Green Etxabe et al.
Copyright year: 2015
Copyright holder: Green Etxabe et al.
License: This is an open access article distributed under the terms of the Creative Commons Attribution License, which permits unrestricted use, distribution, reproduction and adaptation in any medium and for any purpose provided that it is properly attributed. For attribution, the original author(s), title, publication source (PeerJ) and either DOI or URL of the article must be cited.
License URL: https://creativecommons.org/licenses/by/4.0/

Keywords: Amphipoda, Crustacea, Invasive species, Intersexuality, Microsporidia

Funding: Environmental agency University of Portsmouth Strategic Research Funding was provided by the Environmental agency (UK) and a University of Portsmouth Strategic Research Grant. The funders had no role in study design, data collection and analysis, decision to publish, or preparation of the manuscript.

==============================
Crustacean intersexuality is widespread and often linked to infection by sex-distorting parasites. However, unlike vertebrate intersexuality, its association with sexual dysfunction is unclear and remains a matter of debate. The ‘Demon Shrimp,’ Dikerogammarus haemobaphes, an amphipod that has invaded continental waterways, has recently become widespread in Britain. Intersexuality has been noted in D. haemobaphes but not investigated further. We hypothesise that a successful invasive population should not display a high prevalence of intersexuality if this condition represents a truly dysfunctional phenotype. In addition, experiments have indicated that particular parasite burdens in amphipods may facilitate invasions. The rapid and ongoing invasion of British waterways represents an opportunity to determine whether these hypotheses are consistent with field observations. This study investigates the parasites and sexual phenotypes of D. haemobaphes in British waterways, characterising parasite burdens using molecular screening, and makes comparisons with the threatened Gammarus pulex natives. We reveal that invasive and native populations have distinct parasitic profiles, suggesting the loss of G. pulex may have parasite-mediated eco-system impacts. Furthermore, the parasite burdens are consistent with those previously proposed to facilitate biological invasions. Our study also indicates that while no intersexuality occurs in the native G. pulex, approximately 50% of D. haemobaphes males present pronounced intersexuality associated with infection by the microsporidian Dictyocoela berillonum. This unambiguously successful invasive population presents, to our knowledge, the highest reported prevalence of male intersexuality. This is the clearest evidence to date that such intersexuality does not represent a form of debilitating sexual dysfunction that negatively impacts amphipod populations.

Introduction

Dikerogammarus haemobaphes (Eichwald, 1841), an effective predatory amphipod from the Ponto-Caspian (Bacela-Spychalska & Van Der Velde, 2013), has spread through Europe and is now recognised as an extremely successful invader of British waterways (Green Etxabe & Ford, 2014). D. haemobaphes, also known as the ‘demon shrimp,’ invaded the British Isles more recently than the infamous ‘killer shrimp’ (Dikerogammarus villosus, Sowinsky, 1894) (Macneil et al., 2010) but is already more widespread (Fig. 1). Amphipods harbour many parasites that can drastically impact host populations by influencing the health, behaviour, reproduction and sex determination of their host (Hatcher & Dunn, 2011; Bacela-Spychalska et al., 2012). The invasive D. haemobaphes, therefore, could not only outcompete and prey on native amphipod species, but also introduce parasites into their new habitats. Screening parasites in invasive and native amphipod species associated with a rapid and on-going invasion will test hypotheses that particular parasitic burdens impact invasion success MacNeil et al., 2003a; Hatcher & Dunn, 2011; Hatcher, Dick & Dunn, 2014.

Figure 1 Recent confirmed reports of D. haemobaphes (green triangles) and D. villosus (red circles) in UK waterways (EA–unpublished data January 2014; image courtesy of SE Environment Agency).

Some amphipod-infecting parasites maximise their transmission via the host’s progeny by converting males into reproductive females (Ford, 2012). Infection by such parasites results in sex-biased populations (Terry et al., 2004) and, in cases of incomplete conversion, intersexuality, where individuals present secondary sex characteristics of both genders (Ford, 2012). Intersex phenotypes are found in a range of animals (Matthiessen & Gibbs, 1998; Harris et al., 2011; Hayes et al., 2002), including crustaceans (Ginsburger-Vogel, 1991; Bishop, 1974; Ford, 2012), where they are linked to parasitic infection (Li, 2002; Short et al., 2012a) and environmental conditions (Dunn, Mccabez & Adams, 1996), as well the direct (Short et al., 2012b) and indirect (Jacobson et al., 2010) influence of contaminant exposure. In cases of parasitic infection, an incomplete conversion is thought to occur due to insufficient parasite burden, suboptimal conditions, or effective host responses (Dunn & Rigaud, 1998; Kelly, Dunn & Hatcher, 2002; Short et al., 2014). Current evidence suggests that the impact of female intersexuality is subtle (Ford et al., 2003; Kelly, Hatcher & Dunn, 2004) or effectively non-existent (Glazier, Brown & Ford, 2012), and that the female intersexuality observed in D. haemobaphes successfully invading Polish waterways (Bacela, Konopacka & Grabowski, 2009) is consistent with these hypotheses.

Male intersexuality is widespread in amphipods; however, our understanding of its reproductive consequences is poorly understood relative to vertebrates (Harris et al., 2011). The extents of morphological and behavioural changes (McCurdy et al., 2008; Yang, Kille & Ford, 2008) have led to the suggestion (Yang, Kille & Ford, 2008; Ford, 2012) that the impact of crustacean male intersexuality may be similar to that seen in vertebrates (Harris et al., 2011). Despite some evidence of intersexuality in invasive D. haemobaphes (Bacela, Konopacka & Grabowski, 2009), sexual phenotypes in this species have not been studied, even though notable levels of intersexuality in the unambiguously successful invading population would reveal considerable insight into the consequences of intersexuality for wild crustacean populations.

This study investigates the sexual phenotypes and parasites of D. haemobaphes and the native Gammarus pulex (Linnaeus, 1758) at multiple locations in British waterways to give insights into this rapidly invading species and expand our understanding of crustacean intersexuality.

Methods

Specimen characterisation

Amphipods were collected from Wallingford Bridge and Bell Weir, U.K. Amphipods were categorised into species and phenotypes: males, females, intersex males and intersex females. Intersex males were identified by genital papillae, between pereonite 7 and pleonite 1, in conjunction with rudimentary oostegites. Intersex females were identified by oostegites in conjunction with secondary genital papilla/e. Animals from each phenotype were measured from antennal joint to telson to obtain body length (ImageJ, v1.4u4) and comparisons were made using analysis of variance (ANOVA) with the post hoc Tamhanes-T2 test (SPSS v21).

Scanning electron microscopy

Specimens of D. haemobaphes were taken through transitional steps (100% ethanol to 100% hexamethyldisilazane, HMDS) then evaporated to dryness. The dry samples were mounted on SEM stubs, sputter-coated with gold-palladium and examined using a scanning electron microscope (JSM-606LV; JEOL, Welwyn Garden City, Herts, UK) operating in high vacuum mode with a secondary electron detector active at an acceleration voltage of 10 kV. Images were cropped and colourised using Adobe Photoshop (CS5v12).

PCR screen

DNA was purified from internal animal tissue (excluding gut) or eggs using the DNeasy Blood and Tissue Kit (Qiagen, North Manchester, UK). Samples were screened using previously described PCR primers for general parasites (Table 1). PCR reactions were performed in 25 µl volumes containing 10 ng of DNA as template, 1 U of Taq polymerase (Promega, Southampton, Hampshire, UK), 5 µl of 5× PCR buffer, 1.25 mM MgCl2 and 0.4 mM of each corresponding primer. Quality of the DNA samples were analysed using the primers 1073F and 18SR (Table 1) which amplified a 867bp product of the host 18S ribosomal RNA gene.

Table 1 Primers used to conduct parasite screen.

Target	Primer	Sequence	Reference	
Microsporidea 16S	V1f	5′-CACCAGGTTGATTCTGCCTGAC-3′	Weiss et al., 1994	
	1342AC	5′-ACGGGCGGTGTGTACAAGGTACAG-3′	Yang et al., 2011	
Acanthocephala 18S	537F	5′-GCCGCGGTAATTCCAGCTC-3′	Near, Garey & Nadler, 1998	
	1133R	5′-CTGGTGTGCCCCTCCGTC-3′		
	1073F	5′-CGGGGGGAGTATGGTTGC-3′		
	18SR	5′-TGATCCTTCTGCAGGTTCACCTAC-3′		
	18SF	5′-AGATTAAGCCATGCATGCGTAAG-3′		
	549R	5′-GAATTACCGCGGCTGCTGG-3′		
Nematode/acanthocephala/apicomplexa	Nem18SlongF	5′-CAGGGCAAGTCTGGTGCCAGCAGC-3′	Wood et al., 2013	
	Nem18SlongR	5′-GACTTTCGTTCTTGATTAATGAA-3′		
Paramyxea	Para18SF3	5′-CTACGGCGATGGCAGGTC-3′	Short et al., 2012b	
	Para18SR3	5′-GGGCGGTGTGTACAAAGG-3′		
Wolbachia	WSPEC-F	5′-CATACCTATTCGAAGGGATAG-3′	Werren & Windsor, 2000	
	WSPEC-R	5′-AGCTTCGAGTGAAACCAATTC-3′		

Sequence identification

PCR products were analysed using agarose gel electrophoresis containing 1x GelGreen™ (Cambridge Bioscience, UK) for the presence of bands potentially representing amplified parasite sequences. Individual bands were isolated and DNA extracted using the QIAquick Gel Extraction Kit (Qiagen, North Manchester, UK) and sequenced (Source Bioscience, Cambridge, UK), before a BLAST analysis was performed against sequences stored in GenBank (NCBI).

Results

Sexual phenotypes

Pronounced male intersex phenotypes were found in D. haemobaphes at both sites, with most specimens displaying well-developed oostegites with visible setae (Fig. 2). Almost half the male population presented intersex characteristics at both locations and very few cases of female intersexuality were observed (Fig. 3A). G. pulex was only found in conjunction with D. haemobaphes at one sampling site and no intersex phenotypes were found (Fig. 3A). Significant differences were found in lengths of D. haemobaphes phenotypes (F = 3.885, df = 2, p = 0.023) where normal males (14.85 mm ± 3.65, N = 32) are significantly larger (p = 0.04) than females (13.01 mm ± 2.29, N = 52). However, there is no significant difference between intersex males (13.57 mm ± 3.10, N = 37) and either females (p = 0.735) or males (p = 0.328), therefore forming an intermediate size.

Figure 2 External sexual phenotypes.

(A) Normal female D. haemobaphes with only oostegites (green). (B) Intersex male D. haemobaphes specimen presenting genital papillae (purple) alongside oostegites (green) with rudimentary setae. (C) Normal male D. haemobaphes with only genital papillae (purple).

Parasite screening

Screening of D. haemobaphes and G. pulex populations revealed evidence of infection by several parasites (Table 2). All D. haemobaphes females and intersex males were found infected with D. berillonum, with one female weakly infected (Fig. 3B), as previous defined (Yang et al., 2011). The majority of males were also infected, although more weak infections were found (Fig. 3B). This pattern of D. berillonum infection was consistent at both collection sites and when combined in a Fishers Exact test (two-tailed) reveal a significant difference in the level of infection between normal and intersex males (p = 0.003 using strong infections only, p = 0.02, using weak and strong infections). To confirm vertical transmission, the broods of ten infected females were also tested and all were infected by D. berillonum. Only one case of weak D. berillonum infection was found in G. pulex (Fig. 3B).

Figure 3 Frequency of sexual phenotypes and prevalence of D. berillonum infection.

(A) Sexual phenotypes found in two D. haemobaphes populations and G. pulex. (B) Infection of D. berillonum found in D. haemobaphes and G. pulex found in both sites (NF, Normal Female; EIF, External Intersex Female; NM, Normal Male; EIM, External Intersex Male).

Table 2 A screen of parasites using a subsample of the D. haemobaphes and G. pulex populations revealed infection by a variety of parasites.

Strong infection as defined by previous studies (Yang et al., 2011).

Amphipod	Phylum of isolated parasite	Number of strongly infected animals	Length of ribosomal sequence	Primers used for amplification	GenBank accession of isolated sequence	Closest identity using a BLAST	GenBank accession of closest match	% Identity	
D. haemobaphes	Nematoda	11/60	472bp	537F & 1133R	KM486061	Hysterothylacium deardorffoverstreetorum	JF718550	100%	
	Microsporidia	51/60	1148bp	V1f & 1342AC	KM486059	Dictyocoela berillonum	KF830272	99.9%	
G. pulex	Acanthocephala	3/20	547bp	537F & 1133R	KM486063	Echinorhynchus gadi
Echinorhynchus truttae	AY830156	98%	
	Microsporidia	10/20	1135bp	V1f & 1342AC	KM486060	Pleistrophora hippoglossoideos
Pleistrophora typicalis
Pleistrophora mulleri	EF119339	99.6%	
	Apicomplexa	10/20	402bp	537F & 1133R	KM486064	Mattesia geminate	AY334568	90.2%	

Discussion

Our screen of invasive and native species associated with an extremely successful, and ongoing, amphipod invasion reveals parasitic-profiles strikingly consistent with hypotheses that particular parasitic burdens influence the dynamics of biological invasion (MacNeil et al., 2003a; Hatcher & Dunn, 2011; Hatcher, Dick & Dunn, 2014). The native G. pulex are infected with a microsporidian of the genus Pleistophora, which include behaviour-altering species known to increase the likelihood of predation on native amphipods and reduce their predatory behaviour when interacting with invaders (MacNeil et al., 2003a; Fielding et al., 2005). Sequences were also found for an acanthocephalan, most likely Echinorhynchus truttae. This species can both reduce its host’s predatory behaviour and increase vulnerability to predation by fish (Fielding et al., 2003; MacNeil et al., 2003b; Lagrue, Güvenatam & Bollache, 2013). Consequently, the parasite burden of G. pulex may facilitate invasion of D. haemobaphes through British waterways by impairing the competitive abilities of the native population, a scenario consistent with recent experiments and population modelling (MacNeil et al., 2003a; Haddaway et al., 2012; Hatcher, Dick & Dunn, 2014). In contrast, the invasive D. haemobaphes was almost ubiquitously infected by the vertically transmitted microsporidian Dictyocoela berillonum. It is possible the initial invasive population consisted of a small number of infected individuals and the current infection prevalence represents a parasitic founder-effect. Alternatively, given that parasite infection is predicted to influence invasion success (MacNeil et al., 2003a; Fielding et al., 2005; Hatcher, Dick & Dunn, 2014) via trait-mediated effects, it is possible the high prevalence of D. berillonum occurs due to a subsequent enhancement in invasive capabilities.

The distinct parasitic profiles of G. pulex and D. haemobaphes may have ecological impacts. Our results suggest the eradication of native G. pulex would lead to the removal of a pleistophoran microsporidian from the ecosystem potentially capable of causing disease in fish (Lom & Nilsen, 2003) and an acanthocephalan indistinguishable from E. truttae (García-Varela & Nadler, 2005). Although E. truttae infection in fish does not appear to cause morbidity (Dorucu et al., 1995), infected amphipods are more vulnerable to fish predation due to altered habitat usage (MacNeil et al., 2003b; Lagrue, Güvenatam & Bollache, 2013). Therefore, the loss of this parasite may alter prey abundance, even if the overall amphipod biomass is maintained following the displacement of G. pulex.

The sexual phenotype survey revealed that while no intersexuality was evident in G. pulex, the invasive D. haemobaphes presents striking levels of pronounced male intersexuality, where males exhibit unambiguous oostegites possessing rudimentary seta, and their size is not significantly different from males or females. In contrast, the low levels of female intersexuality in D. haemobaphes were much like those previously reported in Polish waters (Bacela, Konopacka & Grabowski, 2009). To our knowledge, this is the highest prevalence of male intersexuality recorded in an amphipod population (McCurdy et al., 2004; Ford & Fernandes, 2005; Short et al., 2012b; Yang et al., 2011) and is the first evidence clearly linking D. berillonum with amphipod intersexuality (Terry et al., 2004; Yang et al., 2011). Other Dictyocoela species have been linked to both abnormal sexual phenotypes and female-biased sex ratios (Terry et al., 2004; Short et al., 2012a), however, the lack of female-bias in D. haemobaphes suggests D. berillonum is unable to fully convert males in to females. This could result from sub-optimal environmental conditions impacting the efficacy of conversion or the consequence of D. berillonum infecting an unfamiliar host. Whatever the cause, the D. haemobaphes intersexuality is of interest. The association between male intersexuality and sexual dysfunction is, despite recent molecular advances (Short et al., 2014), still poorly understood. The functional impact of D. haemobaphes intersexuality is unclear but must incur some form of cost, even if the production of non-functional oostegites on intersexes is merely reducing the resources available for normal growth and reproduction. It is also possible that intersexuality is the outward manifestation of more serious sexual dysfunction. Lower sperm counts have been reported in intersex males of Echinogammarus marinus (Yang, Kille & Ford, 2008) and in Corophium volutator, also females mating with intersex males produce smaller broods (McCurdy et al., 2004). Furthermore, intersexuality may be associated with behavioural changes. Gammarid amphipods mate after a period of mate-guarding and a reduced capacity of intersex males to initiate or maintain this behaviour could also impact reproductive success. The plausibility of such altered behaviours is made more likely given numerous behavioural changes observed in C. volutator intersexes (McCurdy et al., 2008). Investigation of D. haemobaphes reproductive function and behaviour will help determine the extent of dysfunction associated with the intersexuality.

Although the observed intersexuality will incur some cost, the fact that such high levels of pronounced intersexuality has not impeded a successful amphipod invasion is the strongest evidence to date that crustacean male intersexuality is not, in any meaningful sense, equivalent to vertebrate male intersexuality, which is commonly associated with serious sexual dysfunction (Jobling et al., 1998; Harris et al., 2011; Kidd et al., 2007). Furthermore, our findings are consistent with experimentally generated hypotheses that certain parasitic burdens facilitate biological invasions.

Additional Information and Declarations

Competing Interests

Author Contributions

DNA Deposition

ATF is an Academic Editor for PeerJ. Tim Flood and Tim Johns are employees of the Environment Agency (UK).

Amaia Green Etxabe, Stephen Short and Alex T. Ford conceived and designed the experiments, performed the experiments, analyzed the data, contributed reagents/materials/analysis tools, wrote the paper, prepared figures and/or tables, reviewed drafts of the paper.

Tim Flood and Tim Johns analyzed the data, contributed reagents/materials/analysis tools, wrote the paper, prepared figures and/or tables, reviewed drafts of the paper.

The following information was supplied regarding the deposition of DNA sequences:

KM486061

KM486059

KM486063

KM486060

KM486064.

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
