# Peer review of "Pronounced and prevalent intersexuality does not impede the ‘Demon Shrimp’ invasion"

_PeerJ, doi:10.7717/peerj.757_

## Round 0.1 · original submission · Minor Revisions

· Academic Editor

Minor Revisions

The influence of parasitism on intersexuality and how it might influence invading population is an interesting and still understudied research topic. The manuscript is in a good state (see comments by the reviewers), but there all still points which needs to be addressed before it can be published in PeerJ.

The main points to be addressed are:

1) Merge categories “weak” and “strong infection”: You semi-quantitatively estimate the intensity of the micrisporidian infection following Yang et al. 2011 by variation in PCR band intensity which is dependent on various factors not necessarily related with the intensity of the infection (see comments by reviewer 2). Please merge the categories of weak and strong infections as this seems to have no effects on the main results and discussion.
2) The inconsistent and improper use of “significant” and p-values: As also pointed out by reviewer 2, the use of significance in the discussion and conclusions, is not inline with the listed p-values. Please verify these values and correct where necessary (see particularly comments by reviewer 2). It would help to list significant values in bold.
3) Are male intersexes functional or not ? As pointed out by reviewer 2, it might be important to investigate of male intersexes are functional or not as they do not seem to impede invasion (see comments of reviewers 2). Unsure, it these observations suggested by reviewer 2 can still be made, but it is well worth considering to make the paper even more interesting.
4) Please cite list all authors in the text when citing references with 3 authors (see comments by reviewer 1)

In addition, to the points raised by the reviewers, please also address the following points:

Line 60: please replace “occur on a single” with “within a single”

Line 62: The paper might benefit from discussing the occurrence of intersexuality and its link with parasitism more generally as opposed to that in vertebrates. Interesting, intersexuality potentially caused by barnacles can sometimes also be recognized in fossil crustaceans (e.g., Feldmann 1998: www.jstor.org/discover/1306647).

Line 74: it is unclear to what “also” refers so I would just delete it

Line 117: something might have gone wrong with the pdf creation. Please verify which symbol the box behind “GelGreen” should represent.

Line 133: the p-values are not significant; did you list the wrong values here; you need to be more clear what values are significant by marking them in bold; others should be discussed as non-significant

Line 144: please list significant p-values in bold; please merge the strong infections and weak infections together as suggested also by reviewer 2

Line 180: “infected amphipods”: also the invaders ?

Line 186, 194: you list which taxa are invading and which taxa have been living in this region since a longer time in the beginning, but it might help the reader to repeat this from time to time. Potentially you could speak about “invasive” or native in front of the taxon or behind it in brackets.

Line 207-208: Please list some references for “serious sexual dysfunction”

Reviewer 1 ·

Basic reporting

1. The results are the separate chapter, however I agree that in this case it helps to follow the text.
2. citation of the references with 3 authors in the text: in the msc it is cited: 1st author name et al and according to the PeerJ rules all three names should be given.

Experimental design

No Comments

Validity of the findings

No Comments

Additional comments

No Comments

·

Basic reporting

This manuscript describe both high prevalence of intersexuality and parasite infection (by vertically –transmitted microsporidia) in the invasive Gammarid species Dikerogammarus haemobaphes, a new colonizer of Britain. Other, less prevalent, parasites were also identified. The involvement of these infections on the invasion success are discussed, as well as the impact of intersexuality.
It is clear from the results that, while intersexes are present at incredible rate, this did not impede invasion. I would suggest authors to discuss a bit further this point. Apparently, there are no data investigating if these “male intersexes” are functional males or not. Gammarids mate after a period of mate guarding, and this trait could be quite useful to estimate if intersexes are functional or not (at least able to initiate guarding or not). During animal collection, did authors noted if animals (especially male intersexes) were in pair or free? This could provide a clue for intersexes sexual activity. Even if they are not functionally able to inseminate females, if they are able to initiate mate guarding, they may influence mating dynamics by making females with which they are paired unavailable for “real” males…

Experimental design

The intensity of the microsporidian infection has been estimated (heavy infection vs. weak infection), based on a method published previously by the same research group (Yang et al. 2011, see reference list). By looking at this reference, it appears that this semi-quantification is based on the band intensity after microsporidian amplification by classical PCR (not quantitative). No replicate has been made, no DNA quantification before amplification is provided. Variation in PCR band intensity is highly condition dependent (extractions are made individual by individual, so DNA quantity or quality differ between individuals) and often non repeatable. This method is therefore not reliable. The category “weak infection” should be removed and included in the “infection” category. There would be no impact of this deletion on the overall discussion.

Validity of the findings

The discussion/conclusions about size differences between intersexes and “normal” sexes are quite disconnected with the statistics. L. 130-131, “significant differences” are claimed, while p = 0.23. I suppose the overall model is NS, but post-hos test provide a marginal significance for difference between males and females (L. 132). Post-hoc tests cannot be taken into account if the general model is NS. More importantly for conclusions, L. 188, male intersexes are said to be “significantly smaller than normal males”, while stats L. 133-134 provide NS differences. All this should be corrected.

Additional comments

While I think these data deserve to be presented, I have two concerns with this manuscript in its present form: see "experimental design" and "Validity of findings"

---

## Round 0.2 · Minor Revisions

· Academic Editor

Minor Revisions

Thanks for integrating our suggestions and responding to the questions of the reviewers and myself. The manuscript is as good as accepted. Just one minor thing which still needs to be implemented:
Reference "Green Etxabe & Ford 2014" mentioned on line 36 is missing from the reference list. Please integrate this reference now as we can´t (easily) make this change after the manuscript is accepted.

---

## Round 0.3 · accepted · Accept

· Academic Editor

Accept

Thanks for adding this final reference. Your paper is hereby accepted for publication. Congratulations and thanks for submitting this interesting work to PeerJ.